# Behind the Land Use Mix: Measuring the Functional Compatibility in Urban and Sub-Urban Areas of China

Haochen Shi [1,*], Miaoxi Zhao [1], Duncan A. Simth [2] and Bin Chi [2]

1 State Key Laboratory of Subtropical Building Science, School of Architecture, South China University of Technology, Guangzhou 510640, China; arzhao@scut.edu.cn
2 Centre for Advanced Spatial Analysis, The Bartlett Faculty of the Built Environment, University College London, London WC1E 6BT, UK; duncan.a.smith@ucl.ac.uk (D.A.S.); bin.chi.16@ucl.ac.uk (B.C.)
* Correspondence: ar_shc@mail.scut.edu.cn

**Abstract:** Land use mix (LUM) has long been employed as one of the key methods to improve urban vibrancy and optimize built-up areas. Within the urban studies discipline, LUM is usually defined as a functional compatible but diverse land use pattern. However, its quantitative methodological approaches thereby heavily rely on the diversity of land use and fail to consider functional compatibility as another critical defining characteristic, providing only a partial picture of land use pattern. Thus, reviewing LUM's concepts and definitions, this paper develops a new index to describe functional compatibility according to the spatial segregation measurements. To evaluate and provide empirical evidence of the proposed index, this paper selects the medium-sized city of Xiangtan as a case study. The findings demonstrate that Xiangtan exhibits a quite compatible land use pattern to a certain extent. In addition, particular clusters with relatively incompatible land use patterns are observed, which are closely linked to a special historical working unit, the 'Danwei' compounds, and a special rural planning authority, 'Township-Village-Enterprise', in China. Finally, an integrated evaluation is conducted based on the proposed index and Shannon entropy index, which can be regarded as a useful tool in future land use planning while contributing to shaping a sustainable form of urban development.

**Keywords:** land use mix (LUM); functional compatibility; parcels; China



## 1. Introduction

Land use mix (LUM) has long been regarded as a heated topic among urban planners. After World War II, with the accelerating urbanization and its consequent restructuring of the socio-economic relations, the complexity of the city has gradually emerged and been noticed by urban scholars [1–5]. They argued that the city is not an assembly of isolated functional components, but a comprehensive and multifunctional entity. Based on theory, many scholars and urban thinkers started to argue that the widespread adoption of the zoning concept with "functional separation" necessarily resulted in a monotonous, isolated urban environment to a certain extent [6]. Addressing these new barriers to designing and planning the built environment, Jane Jacobs in her book The Death and Life of Great American Cities proposed that the built environment, incorporating the mixture of land uses into it, is an essential element of the urban diversity which should be treated as the necessary precondition for enhancing urban vitality [7]. This theory has subsequently inspired a generation of urban theorists and was further adopted by urban planners and policymakers, leading to the development of many famous documents, such as The Charter of Machu Picchu [8], The Charter of the New Urbanism [9], and the Smart Growth Network [10]. The LUM concept called for an integration of different functions, placing an emphasis on greater interaction between different land uses. An area with a high percentage of LUM can be considered as a refined, suitable, and healthy urban form compared to the ordinary [11–14].

However, both Jacobs and New Urbanists implied that the "unconstrained mix" is not always synonymous with "positive". Negative mixed uses of, also defined as functional incompatibility, possibly lead to "repulsion" between two adjacent parcels owing to some objectionable externalities [15,16]. Mixed uses of heavy industry and some environment-sensitive functions (e.g. residence and farmland) are the typical examples for explaining this incompatibility. Many studies have provided evidence of housing values and living conditions that could be immensely impacted by the environmental issues generated by industrial enterprises [17–20]. Quite a few environmentalists also emphasize that the growth of extensive industrial activities, like minerals mining, can cause the accumulation of heavy metal contaminants, and consequently affects their neighboring land, especially farmlands [21,22]. Therefore, when pursuing the LUM, a functional compatible land use pattern should also be involved into the consideration.

Currently, such viewpoint has gained more tractions among Chinese urban researchers. Since the 21st century, China has witnessed a tremendous growth of urbanization, leading to the rise of economic and social development but with alarming implications, including traffic congestion, gentrification, and environmental pollution. Meanwhile, the ever-increasing urban sprawl has been associated with serious interferences of incompatible land uses in the urban and sub-urban areas (e.g. residence and industry), triggering the exacerbation of environmental degradations [23]. To tackle these issues, some scholars have argued that a healthy, reciprocal or, in other words, functional compatible LUM pattern is a possible strategy [15,23,24], since it can achieve Jacob's ideal urban form and effectively improve urban vibrancy. In this context, the central argument of our paper is focusing on functional compatibility, which can be regarded as the critical and necessary components of sustainable urban developments.

Despite the growing significance of LUM in the urban planning field [6,25], research on quantifying the functional compatibility behind LUM is scarce to a certain degree. To fill such a gap and contribute to the theory of LUM, this paper develops a new index to examine the extent to which functional compatibility can be measured. More specifically, this study: (a) gives the definition of functional compatibility through reviewing the existing literature on LUM; (b) introduces an improved quantitative methodology to gauge functional compatibility according to existing measurements; and (c) selects a typical case study, Xiangtan, to provide empirical evidence and evaluate the performance of the proposed method. It is also worthwhile to mention that building on the measurement framework on spatial segregation developed by Wong [26], this paper advances a novel understanding of compatibility in LUM, making it more practical based on a rational approach. In so doing, all the land types are considered in the measurement, particularly the non-built-up land, such as farmlands and forests, which helps to explicitly explore "repulsions" that are imperceptible, e.g., the incompatibility between the heavy industries and farmlands, in the suburban areas.

## 2. Functional Compatibility: An Important but Unreckoned Characteristic behind Land Use Mix

### 2.1. A Definition of Functional Compatibility

To better understand the functional compatibility of LUM, we initially start from the definition of LUM itself. Land use mix (LUM) is referred to as various land uses that coexist in proximity or adjacency of one another [6,15,24,25,27]. One of the essential components of LUM is the degree to which "mix" can be captured and utilized. Many scholars mainly focus on such a topic and attempt to link it with the categories [28,29], intensity [30], and proximity of various land uses [6]. Nevertheless, an important pre-condition of "mix" implied by Jacobs and New Urbanists is ignored to some extent [7]. Jacobs argued that the integration of diverse urban activities can have the potential to improve the urban vibrancy and produce socio-economic benefits, although, in her words, these activities serve as "complementary functions". In other words, only if the mix of land uses satisfies the pre-requisites of reciprocity and diversity can the urban area benefit from this phenomenon.

Thus, except for the "variety" of land use categories, another key component of LUM, functional compatibility is also needed to be regarded. With these considerations in mind, this study, building on Jacobs's ideas [7], defines the functional compatibility of LUM as follows:

'Functional compatibility of LUM is the degree to which various land uses can co-exist and not disturb each other in a pre-defined spatial range.'

Essentially, functional compatibility is to describe the relationships between different land use types. Such relationships (also known as externalities) can to some extent lead to positive (e.g., improvement in the urban vibrancy) [31] or negative (e.g., environmental issues) [32] effects. As Taleai et al. [16] and Willis et al. [33] suggest, addressing externalities is one of the key agendas in regulatory planning procedures while the logic behind it implicitly coincides with the theory proposed in Athens Chapter [34]. Some scholars argue that land usability is to a great extent impacted and constrained by the externalities of their neighboring land uses [16]. Positive externalities are conducive to forming a reciprocal land use pattern, whereas negative externalities often lead to disordered land uses [31,33,35]. Therefore, in an ideal LUM pattern, functional compatibility needs to be carefully considered and explicitly figured out.

### 2.2. Methodologies for Measuring Functional Compatibility

Although LUM is identified with two key characteristics, namely diverse categories of land use and functional compatibility between various land uses, they are understood differently in academic literature. Until now, current methodological approaches focus largely on the former factor and form quite a few identifying methods including Shannon entropy [29,36], dissimilarity index [37], Simpson index [38,39], and many other indices. Song et al. [5] present a comprehensive review on these indices and regard the concepts of "mix" and "diverse land uses" with a high degree of similarity.

However, there are sparse discussions on the measurements of the functional compatibility between different land uses [15,24]. Analogous to the measurement of diversity of land uses, Zhuo et al. concluded that compatibility needs to be measured according to the proximity and quantity of the types of land uses, which can impact the utilization of one another [24]. Taleai et al. developed a GIS-based model for a quantitative assessment of externalities of adjacent land uses at the block scale in Iran [16]. The model was established based on the land use classification of a local context and thereby with limited reference to measuring compatibility in China. Subsequently, Tian et al. put forward the mixed degree index (MDI) to reflect compatible interactions between residential and industrial parcels in peri-urban areas of China [15]. While this method harnesses grid-based fishnet as the fundamental spatial units, it may overlook a few geometric attributes of land parcels. To address this limitation, Zhuo et al. proposed a vector-based mix degree index (VMDI) and a weighted vector-based mix degree index (WVMDI) to comprehensively examine compatibility through vector-based land use data [24]. These latter approaches overcome the loss of topological features caused by gridding, although they fail to integrate all the relevant geometric features (henceforward geo-features) that describe compatibility through one index.

To sum up, as Jacobs suggests, a relatively successful LUM pattern not only contains various land use types, but also minimizes the conflicts between different utilities of land [7]. However, commonly adopted methodologies place an undue emphasis on "diverse land uses", which leads to a sophisticated measurement of diversity while overlooking the functional compatibility dimension with a fragmented measurement [24]. Therefore, to provide a comprehensive understanding of LUM theory, an improved method for better measuring the functional compatibility of different types of land uses needs to be developed.

### 3. An Improved Index for Measuring Functional Compatibility

*3.1. Classifying Land Use Types and Defining Influence Range*

The suitable classification of land use types is the cornerstone of measuring functional compatibility. Land uses in China are currently governed by the Ministry of Natural Resources and depicted by a sophisticated but over-specified land use classification. For instance, the category of "farmland" is divided into three categories, namely rain-fed, paddy, and irrigated. However, these classifications do not meaningfully capture functional compatibility. Therefore, this paper proposes a novel theory-driven classification to set a basis for the current study by integrating and splitting the existing land use categories. Accordingly, land uses are classified as residential land, industrial land, green land, commercial land, public management and service land, transportation land, water, agricultural-related land, municipal utility land, land designated for specific uses, and others. Detailed definitions of these land categories are shown in Table 1.

**Table 1.** The land use types in the research.

| Land Uses | Definition | Land Types |
|---|---|---|
| Residential land | Residential and corresponding services | built-up land |
| Class I industrial land (Class I) | The industry that impacts the residential and public environment to a certain extent | built-up land |
| Class II industrial land (Class II) | The industry that seriously disturbs the residential and public environment. | built-up land |
| Green land | Parks and other public open spaces | built-up land |
| Commercial land | Various types of business activities, catering, hotel, and other services | built-up land |
| Public management and service land (PL) | Administrative, cultural, educational, health and other services, institutions, and facilities | built-up land |
| Transportation land | Urban roads and transportation facilities | built-up land |
| Municipal utilities land (ML) | Supply, environment, and other facilities | built-up land |
| Specially-designated land (SL) | Special purposes such as security | built-up land |
| Water | Rivers, lakes, reservoirs, potholes, ditches, and tidal flats | non-built land |
| Agricultural-related land (AL) | Agricultural-related activities, including farmland, forest, grassland, garden land, and agricultural-related facilities | non-built land |
| Others | Other purposes | non-built land |

In addition to reshaping a system of land use types, it is essential to provide an appropriate, pre-defined spatial range before examining the functional compatibility of land use. There has not been a widely adopted criterion that assigns a geo-entity to such a parameter. Some scholars have developed guidelines drawing on the urban planning field to define the range. For example, based on the concept of the "15-minutes life circle of community" proposed by the Shanghai Government, Zhuo et al. have adopted 1500 meters as a spatial range [24]. Some other researchers drawing from demographics tend to use adjacency rather than a specific range as the spatial basis for demonstrating the influence between different geo-entities [26,40–42]. This concept has also been applied to define spatial range in some well-known geographical indices, such as Local Moran's I [43] and Getis-Ord Gi* [44]. Focusing on measuring the interaction between different land uses, the paper at hand has adopted the adjacency approach to define a spatial range, particularly given the fact that the adjacency between different parcels represents critical geometric features to forming these interactions. More specifically, as shown in Figure 1, if parcel A is adjacent to parcels B, C, D, and E, then all those parcels can be considered within a spatial range influenced by parcel A.

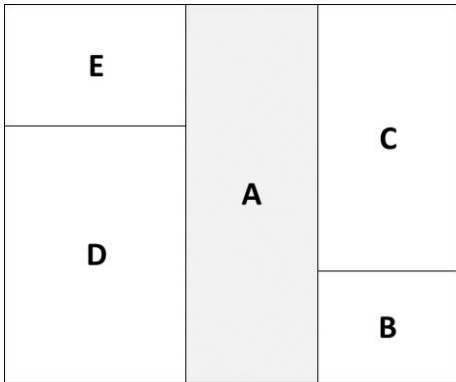

**Figure 1.** Spatial range (including parcels B, C, D, and E) influenced by parcel A.

### 3.2. Measuring Functional Compatibility

The second part is to devise a measurement to capture the functional compatibility. As discussed, the mixed degree index (MDI) proposed by Tian et al. can measure the negative degree of LUM between industrial and residential uses [15], although it only focuses on those two land use types failing to consider all urban activities. In addition, because of its grid-based method, some key geo-features that need to be involved in the calculation process may be lost. The vector-based mix degree index (VMDI) and a weighted vector-based mix degree index (WVMDI) developed by Zhuo et al. can to some extent overcome some of the limitations associated with the use of MDI while extending the scope of measuring compatibility to all land use types [24]. Despite the advantages of VMDI and WVMDI, as suggested by Tobler's first law of geography [45], those indices fall short of measuring compatibility degree regardless of the decaying effects attributed to the distance between parcels.

To overcome the deficiencies of these approaches, this paper proposes a novel index to measure the degree of functional compatibility (henceforward FCDI). In comparison to MDI, VMDI, and WVMDI, this measurement not only considers all the land use types, but also integrates geo-features of parcels examined in the process that are related to compatibility.

The initial step for measuring FCDI is the quantification of compatible relationships between different land use types. In China, quite a few local authorities such as Shanghai have approached FCDI by classifying it into three tiers, compatibility, semi-compatibility, and incompatibility [24,46]. First, compatibility is referred to as adjacent parcels with different land uses that co-exist, benefitting from mutual interactions. Semi-compatibility is identified as neighboring parcels with different land uses that can be mixed only under a certain condition. As for incompatibility, if the adjacent parcels exhibit "disturbed or repelling" characteristics, then those land uses are incompatible. Given this background and based on what Zhuo et al. have developed [24], this paper assigns compatibility as the value of 0, semi-compatibility as the value of 0.5, and incompatibility as the value of 1.0. To provide a more reliable and consistent scoring-based system, a panel of 40 planning experts have been recruited to evaluate the compatibility between different land use types. The average score for each pair is then calculated and inserted into the matrix. Finally, an incompatibility crosstab matrix in Table 2 is determined, where two land uses are compared and assigned a value ranging from 0 to 1. Generally, the higher value is, the more incompatible two land uses are.

**Table 2.** The Land Use Incompatibility Matrix.

| | Residential | PL | Commercial | Class I | Class II | Transportation | ML | Green | SL | AL | Water | Others |
|---|---|---|---|---|---|---|---|---|---|---|---|---|
| Residential | 0 | | | | | | | | | | | |
| PL | 0.113 | 0 | | | | | | | | | | |
| Commercial | 0.138 | 0.013 | 0 | | | | | | | | | |
| Class I | 0.213 | 0.113 | 0.163 | 0 | | | | | | | | |
| Class II | 0.975 | 0.875 | 0.925 | 0.188 | 0 | | | | | | | |
| Transportation | 0.238 | 0.163 | 0.050 | 0 | 0 | 0 | | | | | | |
| ML | 0.850 | 0.575 | 0.538 | — | 0.075 | 0 | 0 | | | | | |
| Green | 0 | 0 | 0 | — | 0.563 | 0 | 0.263 | 0 | | | | |
| SL | 0.575 | 0.775 | 0.813 | — | 0.525 | 0 | 0.488 | 0.063 | 0 | | | |
| AL | 0.063 | 0.063 | 0.063 | 0.113 | 0.463 | 0 | 0.188 | 0 | 0.050 | 0 | | |
| Water | 0 | 0.050 | 0 | 0.088 | 0.613 | 0 | 0.038 | 0 | 0 | 0 | 0 | |
| Others | 0.013 | 0.063 | 0.013 | 0.025 | 0.038 | 0 | 0.125 | 0 | 0 | 0.013 | 0 | 0 |

The definition of each abbreviation is provided in Table 1.

The second step is to identify the key geo-features required for FCDI measurement. The logic behind measuring compatibility is to capture the extent to which the influencing components within a parcel can cross its boundary and impact the adjacent parcels. This logic shares common elements with the theoretical basis of the spatial segregation model proposed by Wong [26], in which three central geo-features determine the interactions between different regions: the length of a shared boundary, area, and perimeter-area ratio (also known as Compactness Index). However, the model based on those geo-features tends to examine the spatially segregated areas globally rather than focusing on a specific parcel. Therefore, this paper has refined some of those features when utilized in the framework. Drawing on Wong's theory and integrating geo-features employed by Tian et al. and Zhuo et al. [15,24,26], the paper's proposed framework considers the length of shared boundaries, the area of adjacent parcels, and the distance between parcels and their neighborhoods. Figures 2 and 3 illustrate the spatial effects of these geo-features in more detail.

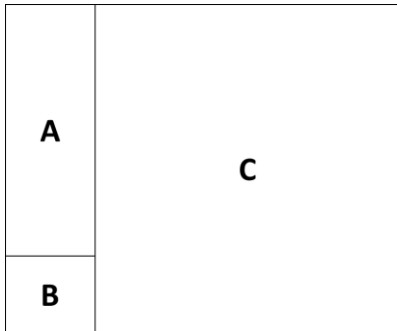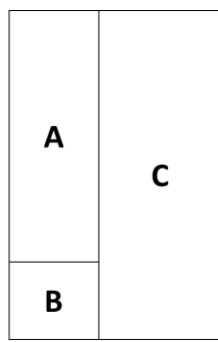

**Figure 2.** Explanation diagram of the geo-features on the length of shared boundaries and the area of adjacent parcels, adapted from the study of Kennedy and Tobler [40]. Assume that the influencing components are evenly distributed in A, B, and C so that C has the most influencing components, while A and B rank 2 and 3, respectively. Given this background, the C influencing components will more easily cross the boundary between C and A than C and B according to the left diagram, as the length of the common boundary between C and A is higher. Similarly, comparing the two diagrams, we can see that C influencing components on the left are obviously greater than C on the right, and thereby the inter influence degree between A and C is larger on the left than on the right.

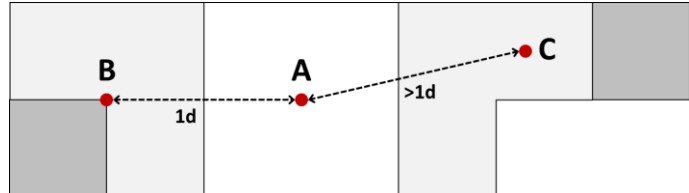

**Figure 3.** Explanation diagram of the geo-features on the distance of centroids between parcels and their neighborhoods, adapted from the study of Massey and Denton [47]. Suppose that: 1.the influencing components are evenly distributed in A, B, and C; 2.the area of A, B and C are same; 3.the length of common boundary between A and B is equal to A and C. In this context, the interaction of influencing components between A and B are greater than between A and C (The light gray area in B and C shares the same area, while the black gray in B is obviously closer to A compared to the area in C). The critical reason for forming such a pattern is that the distance varies between parcels' centroids. In the case, the distance between A and B is 1d and obviously smaller than it between A and C (>1d).

The final step is to quantify the integration of the geo-features as well as the compatibility relationship. Wong used multiplication to integrate the various related geo-components into one index [26], which is similar to the method employed for measuring spatial interactions [48]. One of the essential characteristics of multiplication is that it is sensitive to empty or zero values in its multipliers, which is quite similar to the logic of FCDI measurement. Thus, this study uses multiplication to measure FCDI. It is also necessary to point out some refinements in the measurement process. First, given the importance of a shared boundary in spatial interactions, the distance shown in Figures 2 and 3 is calculated based on the length of the line that connects the parcel centroid to that of the adjacent parcel that they share a boundary (C-C distance). Considering that spatial interaction has a theoretical maximum value rather than infinity, this study has introduced the Gaussian function to describe the distance decay effects.

The specific measurement of FCDI consists of two steps. The first step is to determine the degree of interaction intensity (DI) between the targeted parcel and one of its adjacent parcels. The formula is expressed as follows:

$$DI_{ij} = C_{ij} \times L_{ij} \times A_j \times \frac{1}{\sqrt{2\pi}} e^{\left(-\frac{D_{ij}^2}{2\sigma^2}\right)} \tag{1}$$

where $DI_{ij}$ is the degree of interaction intensity between parcel $i$ and its neighboring parcel $j$. $C_{ij}$ is a value that represents the incompatibility relationship between parcel $i$ and $j$. $L_{ij}$ is the length of the shared boundary between parcel $i$ and $j$. $A_j$ is the area of parcel $j$. $D_{ij}$ is the C-C distance of parcel $i$ and parcel $j$. $\sigma$ is the standard deviation of all the C-C distances in the study area. Due to the quantitative range difference of $C_{ij}$, $L_{ij}$, $A_j$, and $D_{ij}$, all these variables are normalized into [0, 1].

The second step is to calculate the FCDI value of the targeted parcel, which can be expressed as:

$$FCDI_i = 1 - \frac{\sum_j^n DI_{ij}}{max\left(\sum_j^n DI_{ij}\right)} \tag{2}$$

where $n$ is the total number of parcels neighboring parcel i. $DI_{ij}$ is the result of Equation (1). Because the survey in Table 2 is to evaluate the incompatibility relationship, $DI_{ij}$ calculated in Equation (1) stands for the degree of incompatible interaction intensity. Here, a subtraction calculation process is used to reverse the situation. As a result, the higher the FCDI of a parcel, the more compatible the parcel will be with its neighbors. In addition, two small-scale experiments based on our measurements are also conducted to verify the feasibility (See in Figure 4, case A and B).

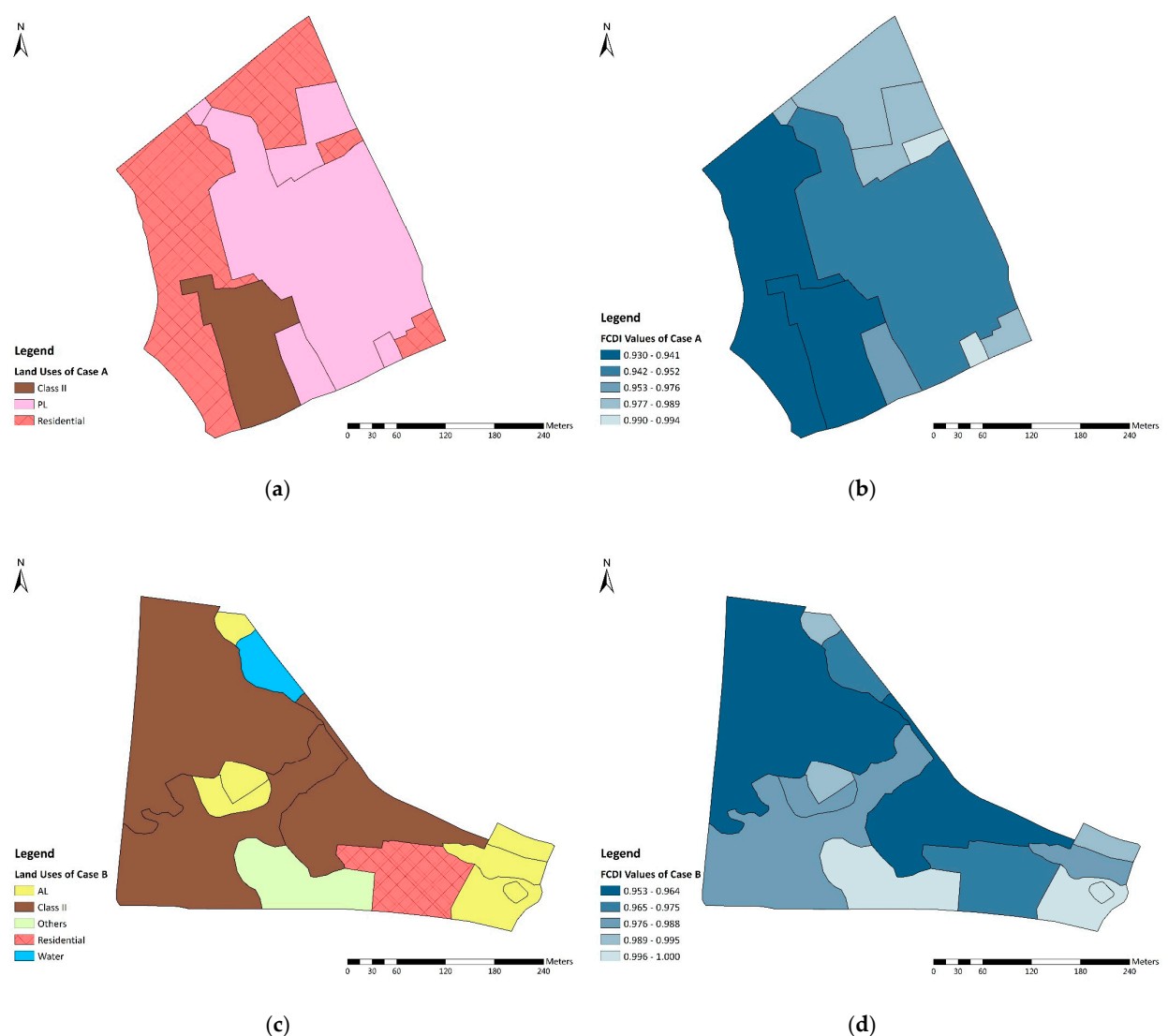

**Figure 4.** Two small-scale experiments of the FCDI index in the study area. (**a**) a land use map of case A. (**b**) the tested FCDI values of case A. (**c**) a land use map of case B. (**d**) the tested FCDI values of case B.

### 3.3. Study Area and the Data

To provide empirical evidence and examine how the proposed index works in practice, this paper now turns to the selected case study of Xiangtan, a medium-sized city located in the southern part of China (Figure 5). Xiangtan is one of the sub-center cities in the Chang-Zhu-Tan City Region with its GDP ranked 123 in 2019 among 293 prefecture-level cities in China. The city's economy is mainly based on industry accounting for 49.3% of the city revenues. The reasons for selecting this case study area are threefold. First, Xiangtan is representative of mid-sized cities in China as economic indicators suggest that Xiangtan's urbanization rate and per capita GDP are close to the national average level. Second, Xiangtan's growth has followed an uneven pattern of development and expansion, which has triggered land use issues particularly in the suburban area, a common characteristic observed in many small and medium-sized cities in China. Third, Xiangtan has historically been an industrial city with some industrial parcels in the old town, which can be a prime candidate to be examined in terms of the negative impacts of the industry on the city. To measure LUM in Xiangtan, this paper has focused on the regions within the urban growth boundary of the municipal district in accord with Xiangtan's spatial planning using the latest land use data (2019).

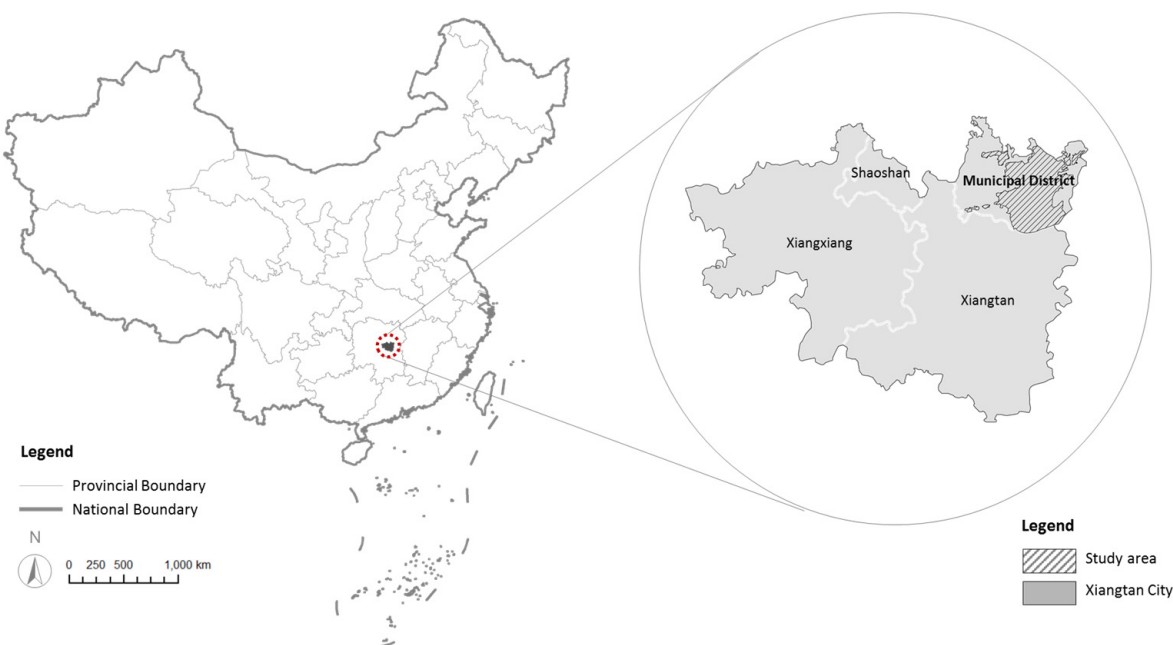

**Figure 5.** The study area, Xiangtan.

## 4. Results and Discussion

### 4.1. Evaluation of Functional Compatibility in the Study Area

Table 3 presents a summary of the descriptive statistics associated with compatibility. Similarly, Figure 6 demonstrates the probability distribution of FCDI with the majority of values falling between [0.9, 1] while the theoretical range is [0, 1]. Such a distribution, on the one hand, indicates that the compatibility of most land parcels in Xiangtan remain at a relatively positive level. On the other hand, it points to the fact that the distribution follows a heavy-tailed one. According to Scripter and Jiang [49,50], to partition and visualize the data using traditional ways, such as Nature Jenks or quantiles, is somewhat inappropriate, as these methods do not provide a clear representation of the data in the long tail. Thus, a new classification, head-tail breaks proposed by Jiang is adopted for providing a greater precision of the targeted data. By implementing the new method, this study has created eight partitioning thresholds and divided all the parcels' FCDI values into seven classes. Details are demonstrated in Figure 6 and Table 3.

**Table 3.** Descriptive statistics of FCDI values of parcels in the study area.

|  | N | Min | Max | Mean | Range |
|---|---|---|---|---|---|
| Class 1 | 24,715 | 0.992 | 1.00 | 0.998 | (0.992, 1.0] |
| Class 2 | 5178 | 0.969 | 0.992 | 0.985 | (0.969, 0.992] |
| Class 3 | 1206 | 0.923 | 0.969 | 0.953 | (0.922, 0.969] |
| Class 4 | 363 | 0.853 | 0.922 | 0.897 | (0.853, 0.922] |
| Class 5 | 117 | 0.754 | 0.853 | 0.816 | (0.754, 0.853] |
| Class 6 | 26 | 0.623 | 0.747 | 0.702 | (0.592, 0.754] |
| Class 7 | 19 | 0.00 | 0.592 | 0.467 | [0.0, 0.592] |
| Total | 31,624 | 0.00 | 1.00 | 0.97 | - |

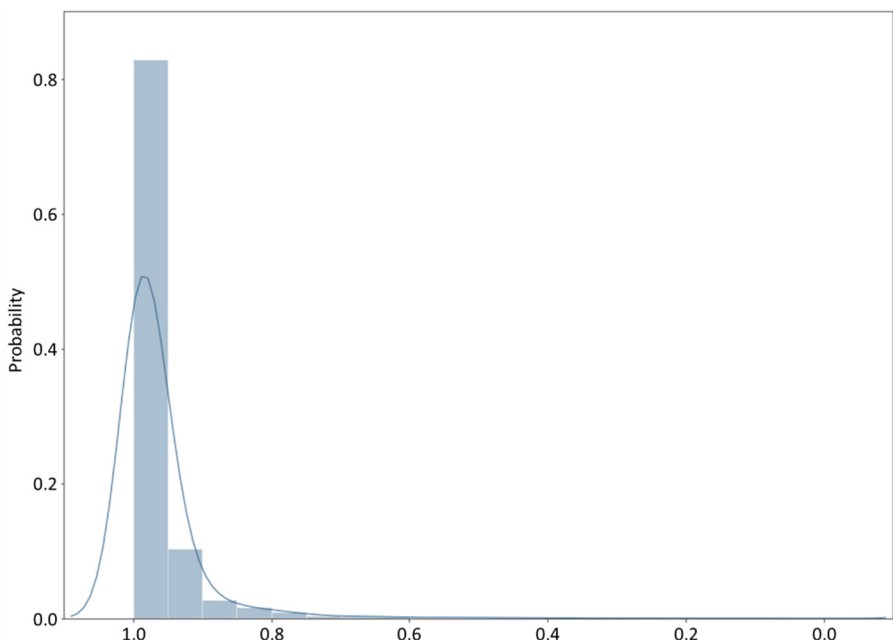

**Figure 6.** The probability distribution of FCDI values of parcels in the study area.

With the new classification, Figure 7 illustrates the spatial distribution pattern of FCDI values in Xiangtan city. Generally speaking, land use patterns in the city center areas of regions A and B show a high level of compatibility with their values falling from 0.8 to 1.0 and from 0.9 to 1.0 respectively. Region C located in the satellite town also shows a relatively compatible pattern of land uses with its FCDI ranging from 0.53 to 1. However, the FCDI of built areas is larger than the non-built areas. In fact, this can be regarded as a natural phenomenon since urban activities in developed areas almost always exhibit a pattern of high intensity, unlike non-built areas. Such intensive activities are closely linked with both positive and negative externalities (e.g., noise) so that the parcels in built-up areas are more likely to impact their neighborhoods compared with non-built regions, which leads to relatively incompatible relationships in built-up areas. This hypothesis can also be supported by the compatibility matrix surveyed from the planning experts presented earlier in Table 2. The incompatible values between built-up land use types range from 0.0 to 0.975, while the same values in non-built areas fall between 0.0 and 0.013.

However, a few clusters with low FCDI values appear in the suburban areas, with the majority of parcels in these incompatible clusters exhibiting characteristics of class II industry (hereafter class II) and partial municipal utilities land uses (hereafter ML). Theoretically, class II and ML are regarded as indispensable functions of cities, often generating spatial externalities, such as noise, wastewater, and air pollution during its working time, leading to huge negative environmental impacts on the surrounding areas. However, these impacts can be effectively minimized through zoning and planning schemes that limited their concentration in certain places, such as industrial parks or districts. In Xiangtan, such zoning practices have occurred but with limited application and only in partial regions such as region C. These clusters, especially the industrial ones, generally lack effective urban management procedures in place leading to the fragmented, intertwined, and incompatible pattern of land uses.

Two reasons particularly stand out when it comes to a lack of comprehensive planning in these clusters. The first one is closely linked to the land management system and property rights in China. Generally, the Chinese land ownership is split into the state and the collective, where the state land, or urban land, is always located in the city center while the collective land usually lies in rural and sub-urban areas. In most cases, the collective land is directly managed by the 'Township-Village-Enterprises' (TVEs) or the administrative village instead of local authorities. In addition, after the Reform Opening, suburban areas

are gradually considered as attractive places for outside industrial investments because of advantageous locations and cheap compensation cost for land requisitions. Giving these two backgrounds, the TVEs have gradually started to establish new industrial clusters on the land they managed as many as possible for acquiring more GDP and profits. During the self-organized industrial development, the only affair that TVEs pay attention to is attracting more investments, while the planning schemes are seldom treated as a key issue as their implementations need to cost plenty of time, effort and monetary investment. Thus, such bottom-up industrializations always suffer from quite a few disorders and a lack of sufficiently scientific management. The direct consequence of this chaotic development is the scattered land use pattern of industry, resulting in highly incompatible mixed uses landscape in suburban areas.

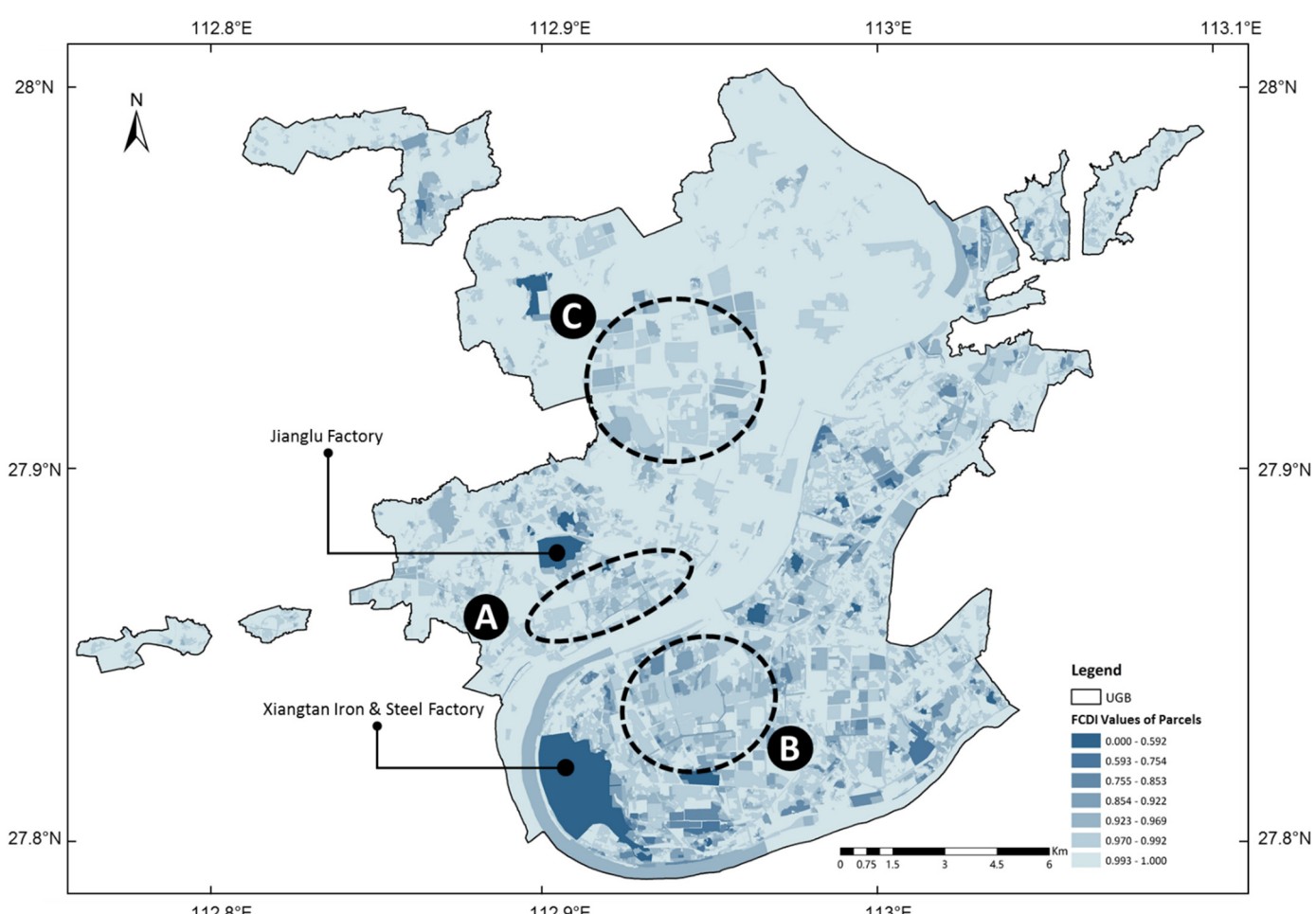

**Figure 7.** The spatial distribution pattern of FCDI values of parcels in the study area.

Another reason is strongly tied with a specially organized model, 'Danwei' compound in the past century. Here, parcels 1 and 2, which used to be the working part of Xiangtan Iron & Steel Factory and the residential part of Jianglu Factory, are the representative examples of such phenomenon. Danwei compounds, also known as work units, have historically been considered as universal but significant neighborhood types before the 1980s. Aiming at reconstructing and redeveloping the local economy, particularly China's industry after 1949, the central government initiated the Danwei system and directly allocated land resources to different state-owned enterprises to establish their own Danwei compounds. These organized communities comprise workplaces, residential areas, and a variety of public amenities, such as canteens, educational facilities, green space, hospitals, and even cinemas. In order to minimize the commute time, various functions are built

within the central region, thus offering a relatively highly interweaved land use pattern. More particularly, owing to the major functions of the 'Danwei' compound in terms of residence and industry, its functional incompatible degree should be relatively higher than a normal community. Since the 1980s, Danwei compounds have gradually been governed by local authorities, and therefore their outdated and unsustainable land use patterns should be reformed and reorganized under the guidance of modern and sophisticated planning schemes. Such redevelopments, also regarded as urban regenerations, would require a large amount of investment and workforce as the land in 'Danwei' compounds has been highly developed. More importantly, these regenerations inevitably involve large physical interventions, such as demolitions and resettlements, often leading to social unrest and socio-economic implications for local communities. Thus, only in some metropolises like Shanghai can regeneration be widely designed and conducted as they not only have enough financial support, but also a mature regeneration mechanism. By comparison, in some small cities like Xiangtan, which lacks both budget and policy support, local authorities are more inclined to develop non-built land in suburban areas as opposed to regenerating Danwei compounds. In other words, incompatible land use patterns in Danwei compounds still exist.

### 4.2. An Integrated Evaluation of LUM Based on FCDI Index

Currently, the urbanization process in China has entered the later stage. Owing to the continuous decrease of farmland and the deterioration of the natural environment, the spatial expansion of urban areas cannot follow the former pattern, which is similar to the early 21st century. The shortage of available land resources for urban development impels local governments to pay more attention to the regenerations and reuses of the developed land with unreasonable uses. Therefore, whether a parcel is functionally incompatible with its neighbors is regarded as a critical characteristic that needs to be identified. In this context, the FCDI we proposed in this paper can provide an improved measurement of the functional compatibility. By introducing the FCDI, planners and urban governors have a new tool for evaluating the degrees of functional compatibility of current land use patterns, which is necessary for subsequent land use planning.

More importantly, as we discussed before, the functional compatibility is a key characteristic of LUM, a practical planning theory of sustainable urban development. Thereby, combining another component of LUM, the diversity of land uses, a comprehensive examination can be conducted and utilized into the evaluation of current land use pattern, giving a quantitative basis and guidance for future land use planning. Herein, as recommended by Song et al. and Yue et al. [6,51], we selected Shannon entropy index (SEI) as the measurement of diversity as it can proportionately consider each land use types. Note that the mixture of various activities bears relevance to an "urban area" [7]. We thus disregard non-built land like cropland and assign its SEI values as 0, as far as diversity is concerned (Detailed measurements of SEI index can be found in Appendix A).

Based on the data from Xiangtan city, bivariate scatter plots are harnessed to classify the corresponding relationships of diversity (SEI index) and functional compatibility (FCDI index) based on four quadrants (Q1, Q2, Q3, and Q4), each of which occupies an area of 0.5 × 0.5. Meanwhile, a distribution map of parcels in Q1, Q2, Q3, and Q4 is also conducted to reveal their spatial characteristics. The details are displayed in Figures 8 and 9. According to them, over half of the parcels belong to Q4 and are more likely to lie in suburban areas. Most of these parcels are utilized as agricultural-related land uses and they have higher FCDI but lower SEI values. The parcels in Q1, with both higher FCDI and SEI values, are mainly located in the city center and the satellite town. They have already formed a relatively rational land use pattern. As for the parcels in Q2, whose FCDI are lower but SEI are higher, it is found that most of them are identified as or used to be Danwei compounds. According to the previous discussion, the highly mixed characteristics in spatial dimension of such unique 'community' naturally tends to form a diverse but functionally incompatible land use pattern. Finally, the parcels in Q3 exhibit low SEI and FCDI values, simultaneously

being scattered in the suburban area while performing industrial functions. A closer look, reveals that most of the parcels are owned by the 'Township-Village-Enterprises' and demonstrate a monolithic, fragmented, and incompatible land use pattern.

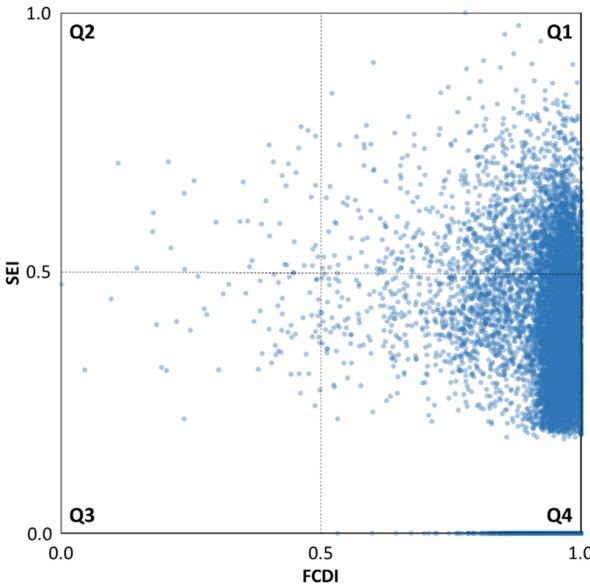

**Figure 8.** Binary scatter plot of SEI (the shannon entropy index) and FCDI (the functional compatibility degree index) values of parcels in the study area.

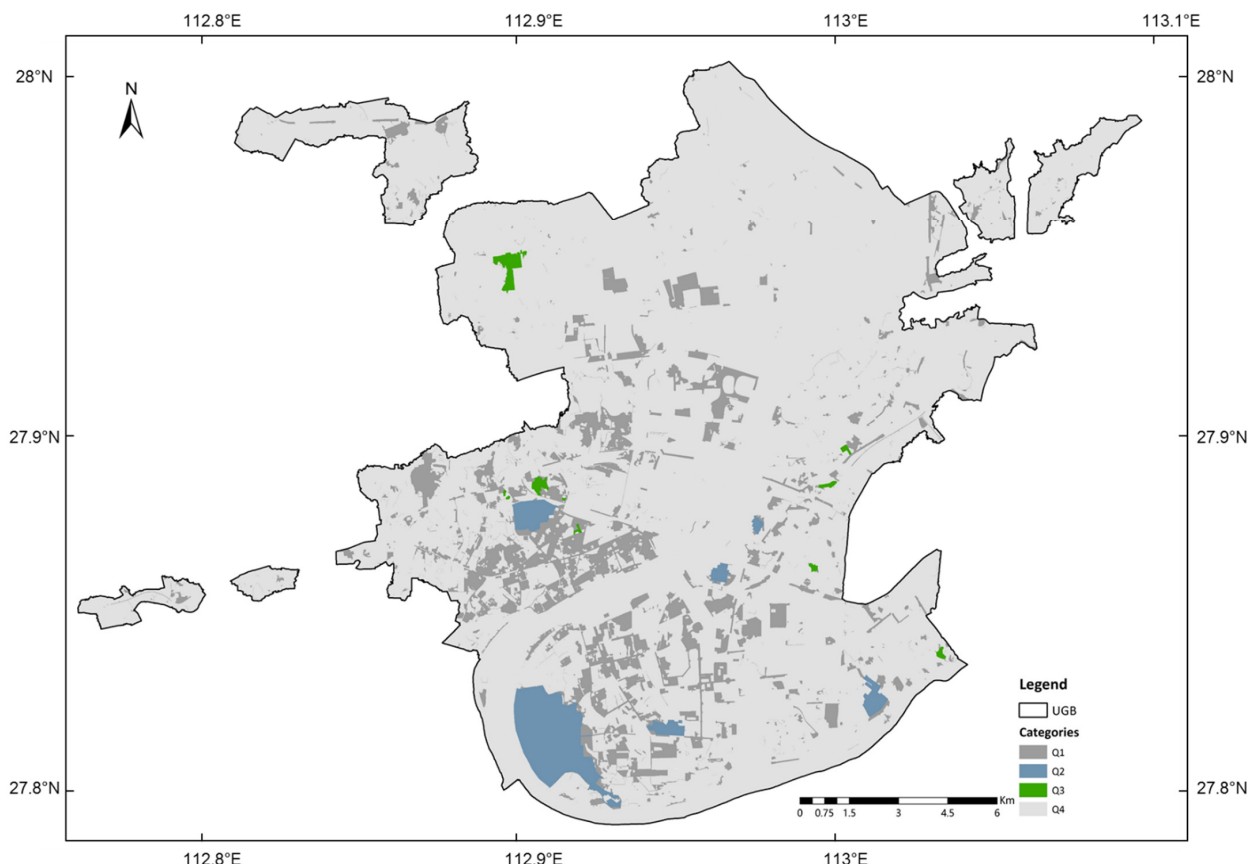

**Figure 9.** The spatial distribution pattern of parcels in Q1, Q2, Q3, and Q4. The detailed definitions of Q1, Q2, Q3, and Q4 are linked to Figure 8.

## 5. Conclusions

Ever since Jacobs brought attention to mixed land use as a balanced mix of working, service, and living activities for a lively, stimulating, and secure public realm in the city [7], the concept of land use mix (LUM) and its significance have become key topics in urban planning theory and practice. Despite LUM being depicted as a functional compatible but diverse land use pattern by Jacobs and New Urbanists, most of its measurements start from the perspectives of diversity of land uses [6,24,25,27,51,52] while ignoring functional compatibility between land uses. Such academic direction has resulted in the triumph of quantitative approaches of the diversity component but relatively scant attention paid to measurements of functional compatibility. Thus, extant literature only provides a partial picture of LUM, possibly leading to imperfect evaluation frameworks of land use patterns.

This paper fills the gap by proposing a novel index, FCDI, to assist in quantifying the functional compatibility in an area where different mixed land uses exist. Starting from the studies of Zhuo et al. and Tian et al. [15,25], the FCDI is developed from the social segregation measurements discussed by Wong [26]. Compared to the previous methodologies, the FCDI first involves all the land use types including non-built ones in order to capture the imperceptible incompatibility. Second, it considers all possible geo-features and includes them into the measurement process. Third, it attempts to use one comprehensive index to reflect the characteristics of functional compatibility.

To provide empirical evidence, this paper has selected a case study of Xiangtan city, a typical medium-sized city in China representative of other third-tier cities to evaluate the proposed index. The general findings suggest that most of Xiangtan's land uses are quite functionally compatible. Although compatible land use patterns tend to be located in either the city center or the suburban area, this is not a universal observation as incompatible land uses do exist, particularly in the suburban regions. The reasons for forming such parcels can be concluded as: (1) the presence of Danwei compounds, the historic concentration of organized working units; and (2) the special land management system and organization 'Township-Village-Enterprise' in China. Additionally, combining another component of LUM, diversity, a map of bivariate scatter plots can be established to address the characteristics of diversity and functional compatibility of current land use patterns, providing urban planners a useful tool for future spatial planning.

In conclusion, our proposed index provides a new methodology to measure functional compatibility, a critical component of LUM. This paper acknowledges a few limitations associated with the study. While the index provides a way to measure compatibility, real-world applications might be limited in situations where there is a certain degree of land use anomalies. Furthermore, measuring compatibility using a quantitative approach is primarily based on the expert panel including planning practitioners, and therefore indigenous and local knowledge could improve the index. Despite the index's novel methodology, certain improvements could make it more refined and precise. Firstly, incorporating the latest research of geography, topology, and other disciplines into the index could be one path to refine it, particularly to improve the compatibility measurement. Introducing some simulation methods such as CA models represents a potential direction for exploring the inner mechanism and improving the measurement of functional compatibility [53–57]. Secondly, adopting mechanisms, such as a group decision making approach that could involve more local planning practitioners and create a more convincing compatibility relation matrix, would be beneficial. Finally, developing an integrated index that can examine the degree of diversity and compatibility simultaneously could be a huge step forward.

**Author Contributions:** Conceptualization, H.S.; methodology, H.S., M.Z. and D.A.S.; software, H.S. and M.Z.; validation, H.S. and M.Z.; formal analysis, B.C.; data curation, H.S.; writing—original draft preparation, H.S.; writing—review and editing, D.A.S. and B.C.; visualization, H.S. All authors have read and agreed to the published version of the manuscript.

**Funding:** This research was funded by National Natural Science Foundation of China (grant number 52178037).

**Data Availability Statement:** Not applicable.

**Conflicts of Interest:** The authors declare no conflict of interests.

## Appendix A

*Appendix A.1. The Measurements SEI (Shannon Entropy Index)*

This section specifically introduces the SEI measurement. Importantly, Jacobs implied the mixture of various activities bears significant relevance in an 'urban area', which can be regarded as built-up land as previously mentioned. Thus, non-built land such as croplands should be disregarded in the calculation process. To handle this issue, we have revisited the Shannon entropy equation and revised some of the prerequisites of land use categories. The specific quantified equations are as follows:

$$A_i = \begin{cases} A_i, i \,\epsilon\, M \\ 0, i \,\epsilon\, N \end{cases} \tag{A1}$$

$$P_i = \frac{A_i}{\sum_{i=1}^{k} A_i} \tag{A2}$$

$$SEI = \begin{cases} -\frac{\left[\sum_{i=1}^{k} P_i \ln P_i\right]}{\ln k} & , P_i \neq 0 \\ 0, P_i = 0 \end{cases} \tag{A3}$$

where $k$ is the number of land use types, $A_i$ is the area of i[th] land use types, while $M$ and $N$ represent the aggregation of built-up and non-built land use types respectively. Generally, a higher SEI value demonstrates greater land use variety, whereas a lower value indicates less diversity. The final spatial distribution of SEI values can be found in Figure A1.

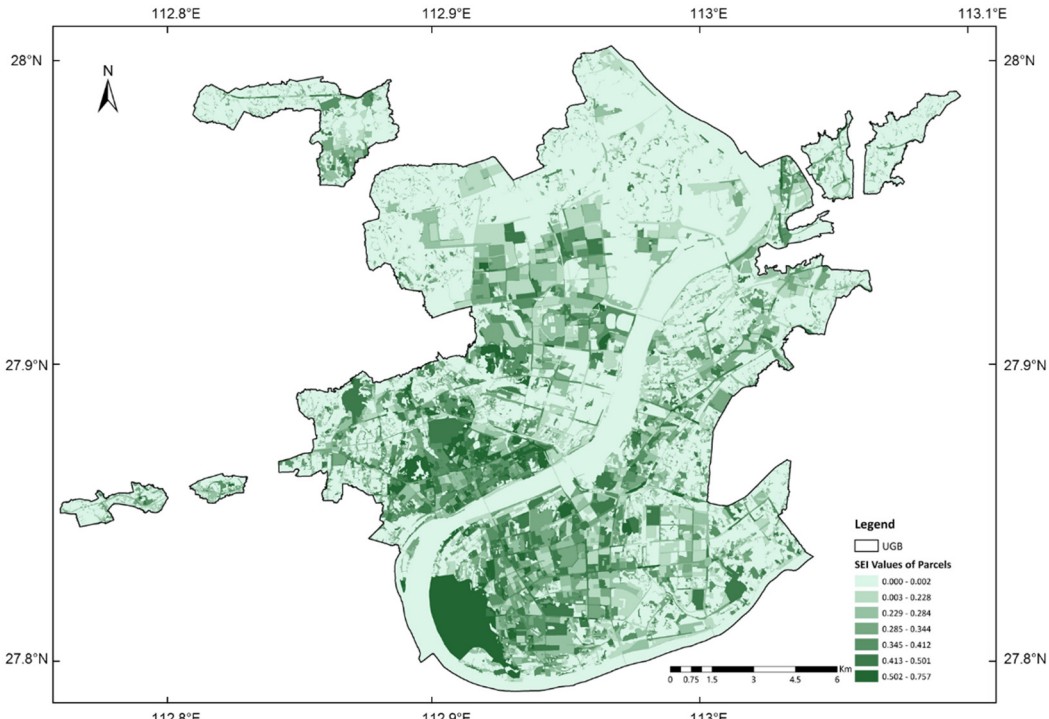

**Figure A1.** The spatial distribution pattern of SEI (the shannon entropy index) values of parcels in the study area.

*Appendix A.2. Detailed Information of the Matrix*

The following table (Table A1) provides the detail information from Table 2. Most of the Std values are smaller than 0.30, while almost all the Average Scores lie closer to the Mode values. These two findings suggest that the Average Scores can represent the opinions of 40 planning experts to a great extent.

**Table A1.** Detailed results of 40 planning experts scores. Relationship between land uses.

| | People Scored 0.0 | People Scored 0.5 | People Scored 1.0 | Average Score | Mode | Std |
|---|---|---|---|---|---|---|
| AL-Transportation | 40 | 0 | 0 | 0 | 0.0 | 0.00 |
| AL-Residential | 35 | 5 | 0 | 0.063 | 0.0 | 0.17 |
| AL-Class II | 5 | 33 | 2 | 0.463 | 0.5 | 0.21 |
| AL-Others | 39 | 1 | 0 | 0.013 | 0.0 | 0.08 |
| AL-SL | 36 | 4 | 0 | 0.050 | 0.0 | 0.15 |
| AL-Water | 40 | 0 | 0 | 0 | 0.0 | 0.00 |
| Others-Water | 40 | 0 | 0 | 0 | 0.0 | 0.00 |
| Others-Residential | 39 | 1 | 0 | 0.013 | 0.0 | 0.08 |
| Transportation-Others | 40 | 0 | 0 | 0 | 0.0 | 0.00 |
| AL-Commercial | 36 | 3 | 1 | 0.063 | 0.0 | 0.20 |
| Green-Transportation | 40 | 0 | 0 | 0 | 0.0 | 0.00 |
| Green-Water | 40 | 0 | 0 | 0 | 0.0 | 0.00 |
| Residential-Water | 40 | 0 | 0 | 0 | 0.0 | 0.00 |
| Residential-Transportation | 35 | 5 | 0 | 0.063 | 0.0 | 0.17 |
| Residential-Class II | 0 | 2 | 38 | 0.975 | 1.0 | 0.11 |
| Residential-ML | 2 | 8 | 30 | 0.850 | 1.0 | 0.28 |
| Commercial-Transportation | 36 | 4 | 0 | 0.050 | 0.0 | 0.15 |
| Residential-SL | 1 | 32 | 7 | 0.575 | 0.5 | 0.21 |
| Class II-Transportation | 40 | 0 | 0 | 0 | 0.0 | 0.00 |
| Class II-Water | 1 | 29 | 10 | 0.613 | 0.5 | 0.24 |
| Class II-ML | 35 | 4 | 1 | 0.075 | 0.0 | 0.21 |
| Transportation-Class I | 40 | 0 | 0 | 0 | 0.0 | 0.00 |
| Others-Class I | 38 | 2 | 0 | 0.025 | 0.0 | 0.11 |
| Class I-AL | 34 | 3 | 3 | 0.113 | 0.0 | 0.28 |
| Transportation-Water | 40 | 0 | 0 | 0 | 0.0 | 0.00 |
| Transportation-ML | 40 | 0 | 0 | 0 | 0.0 | 0.00 |
| Water-ML | 38 | 1 | 1 | 0.038 | 0.0 | 0.17 |
| Water-Class I | 35 | 3 | 2 | 0.088 | 0.0 | 0.25 |
| Transportation-PL | 29 | 9 | 2 | 0.163 | 0.0 | 0.28 |
| Residential-PL | 33 | 5 | 2 | 0.113 | 0.0 | 0.26 |
| Class II-PL | 1 | 8 | 31 | 0.875 | 1.0 | 0.24 |
| AL-PL | 37 | 1 | 2 | 0.063 | 0.0 | 0.23 |
| Residential-Commercial | 29 | 11 | 0 | 0.138 | 0.0 | 0.22 |
| Residential-Class I | 24 | 15 | 1 | 0.213 | 0.0 | 0.27 |
| Water-PL | 36 | 4 | 0 | 0.050 | 0.0 | 0.15 |
| Class I-Class II | 28 | 11 | 2 | 0.188 | 0.0 | 0.29 |
| Green-Residential | 40 | 0 | 0 | 0 | 0.0 | 0.00 |
| Commercial-Others | 39 | 1 | 0 | 0.013 | 0.0 | 0.08 |
| Others-Class II | 38 | 1 | 1 | 0.038 | 0.0 | 0.17 |
| Others-PL | 36 | 3 | 1 | 0.063 | 0.0 | 0.20 |
| Commercial-Water | 40 | 0 | 0 | 0 | 0.0 | 0.00 |
| Green-AL | 40 | 0 | 0 | 0 | 0.0 | 0.00 |
| ML-Others | 32 | 6 | 2 | 0.125 | 0.0 | 0.27 |
| ML-AL | 27 | 11 | 2 | 0.188 | 0.0 | 0.29 |
| PL-Commercia l | 39 | 1 | 0 | 0.013 | 0.0 | 0.08 |
| Commercial-Green | 40 | 0 | 0 | 0 | 0.0 | 0.00 |
| Commercial-Class II | 1 | 4 | 35 | 0.925 | 1.0 | 0.21 |
| Green-Class II | 2 | 31 | 7 | 0.563 | 0.5 | 0.23 |
| ML-Green | 21 | 17 | 2 | 0.263 | 0.0 | 0.30 |
| Commercial-ML | 1 | 33 | 7 | 0.588 | 0.5 | 0.21 |
| SL-Transportation | 40 | 0 | 0 | 0 | 0.0 | 0.00 |
| PL-ML | 2 | 30 | 8 | 0.575 | 0.5 | 0.24 |
| Others-Green | 40 | 0 | 0 | 0 | 0.0 | 0.00 |
| PL-Class I | 33 | 5 | 2 | 0.113 | 0.0 | 0.26 |
| PL-Green | 40 | 0 | 0 | 0 | 0.0 | 0.00 |
| Commercial-Class I | 29 | 9 | 2 | 0.163 | 0.0 | 0.28 |
| Water-SL | 40 | 0 | 0 | 0 | 0.0 | 0.00 |
| Class II-SL | 5 | 28 | 7 | 0.525 | 0.5 | 0.27 |
| SL-Others | 40 | 0 | 0 | 0 | 0.0 | 0.00 |
| SL-Commercial | 3 | 9 | 28 | 0.813 | 1.0 | 0.31 |
| SL-PL | 1 | 16 | 23 | 0.775 | 1.0 | 0.27 |
| ML-SL | 8 | 25 | 7 | 0.488 | 0.5 | 0.31 |
| SL-Green | 36 | 3 | 1 | 0.063 | 0.0 | 0.20 |

The definition of each abbreviation is provided in Table 1.

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
