# Peer review of "Behind the Land Use Mix: Measuring the Functional Compatibility in Urban and Sub-Urban Areas of China"

_land, doi:10.3390/land11010002_

Round 1

Reviewer 1 Report

This paper develops an index of the functional compatibility of different land uses and demonstrates the use of the index with data from Xiangtan, China. The index is referred to as the Functional Compatibility Degree Index (FCDI) and is based in part on a survey of professionals' opinions about the compatibility of various land uses. The calculation of the index also takes into account geographic characteristics of the relevant parcels, including the length of shared boundaries and the areas of the parcels. The index is intended to complement measures of land use mix (LUM) which, according to this paper, neglect to take into account incompatible uses.

I have two major concerns about this paper. The first is that the proponents of LUM, starting with Jane Jacobs and continuing with the New Urbanists and others, never advocated the mixing of clearly incompatible land uses, such as heavy industry with residential uses. This paper incorrectly sets up as a straw man the ideal of mixing uses, regardless of the type of use. New Urbanists, for example, advocate a mix of uses in what many refer to as transect zones (usually labeled T1 through T6), but land uses that generate significant negative externalities are excluded from these zones and located in separate special districts. The idea of including noxious uses in an LUM index does not make any sense. The Shannon Entropy Index (SEI) used in this paper appears to make that mistake.

The second major issue is that the FCDI is probably too crude to serve as a basis for regulating land use. For example, there is undoubtedly a range of industrial uses, some more noxious than others. But the system proposed here has just two classes of industrial uses, which seems too simplistic to adequately measure compatibility or incompatibility. Similarly, uses such as "transportation" are insufficiently precise to serve as a basis for regulation. Moreover, the compatibility matrix in Table 2 is based on professional opinions that may or may not relate closely to real world circumstances. A more specific, performance-based approach to compatibility would probably be a better way to regulate incompatibility. My sense is that the FCDI is potentially useful more as a tool for describing patterns of compatibility and incompatibility than as a basis for land use regulation.

Some smaller points:

  1. The definition of LUM on page 3 seems circular; a better definition is needed.
  2. It would be useful to provide some information about the degree of agreement or disagreement among the 40 planning experts who were surveyed to create the matrix in Table 2. If there is relatively little disagreement, that would make a stronger case for using the matrix.
  3. I could not follow the explanation of the third geographical feature illustrated in Figure 3 on page 7. The reference to "the distance between parcels and their neighborhoods" seems particularly confusing. This needs to be explained more clearly.
  4. Some discussion of the final term on the right-hand side of Equation (2) would help the reader to understand the equation.
  5. I could not follow the discussion in the first paragraph of section 4.1 regarding the distribution of the compatibility measures.
  6. The paper seems to use the terms CDI and FCDI to refer to the same thing. It would be better to consistently use one or the other. If they are two different things, then that needs to be made clear.
  7. What are the units of analysis in Table 3? Are they parcels of land?
  8. In Table 2, higher values seem to be associated with incompatibility while, in Table 3 and Figure 6, lower values seem to be associated with incompatibility. This is quite confusing.
  9. It is not clear what is meant by "values falling between [0.80, 1] and [0.91, 1]" at the top of page 10.
  10. The paper needs to explain exactly how the SEI was calculated. 

Author Response

Dear Reviewer,

Many thanks for the reviewer’s valuable suggestions. We totally agree with comments and concerns put forward by the reviewer, and we carefully revised our manuscript based on these insightful suggestions. It must be pointed out that with the help of the reviewer's comments, the overall structure and the specific expressions of the manuscript have been greatly improved.

The detailed responses and revisions to these suggestions can be found in the response letter and the revised manuscript we uploaded.

Best wishes,

Authors of the Manuscript entitled:

Behind the Land Use Mix: Measuring the Functional Compatibility in Urban and Sub-urban Areas of China

Reviewer 2 Report

This paper is aimed to propose a method to quantify functional compatibility of various land uses in addition to their diversity in the Land Use Mix (LUM) approach of urban planning. The proposed method was evaluated in a case study at China. The English writing of the paper is acceptable. The texts are clear and the paper has been structured well. As a result, I can suggest to publish this paper after minor corrections.

My comments are as follow:

  • I’m not agree with the proposed method for defining “Spatial range” influenced by the parcel, as presented at Fig A. In fact, two parcels placed at two sides of a road and don’t touch, also can have some externalities to each other. Some land use types such as “Industrial” effect other land uses from a far distance. For more information, please have a look on the following papers. You can see how they calculate compatibility of land uses in a parcel based CA models. Maybe you found their proposed approach proper to utilize and integrate to improve your proposed method in this research.
    • Simulating urban growth under planning policies through parcel-based cellular automata (ParCA) model
    • Assessing the effect of temporal dynamics on urban growth simulation: Towards an asynchronous cellular automata
    • Modeling land use interaction using linguistic variables
    • Simulating urban growth in a megalopolitan area using a patch‐based cellular automata
    • Spatial Conflict Reduction in Building Generalization Process using Optimization Approaches
    • An integrated framework to evaluate the equity of urban public facilities using spatial multi-criteria analysis
  • In Table 2, I expect to present more details to describe how the average score was calculated. I think when 40 experts cooperate in this step, a group decision making approach should be utilized to calculate the final score and simple average calculation maybe was unsuitable.
  • The results of the proposed formula to calculate CDI must be tested and verified by doing some experimental activities based on several samples of land use mix of real land use data in the case study area. First, it is necessary to be sure about the proper results of the proposed Equations and then use them to the whole data.
  •  The title of Fig.8 should be corrected
  • I suggest to add a land use map of the case study area.

Author Response

Dear Reviewer,

Many thanks for the reviewer’s valuable suggestions. We totally agree with comments put forward by the reviewer, and we carefully revised our manuscript based on these insightful suggestions. It must be pointed out that with the help of the reviewer's comments, the overall structure and the specific expressions of the manuscript have been greatly improved.

The detailed responses and revisions to these suggestions can be found in the response letter and the revised manuscript we uploaded.

Best wishes,

Authors of the Manuscript entitled:

Behind the Land Use Mix: Measuring the Functional Compatibility in Urban and Sub-urban Areas of China
